# Association between Plasminogen Activator Inhibitor-1 and Osimertinib Tolerance in EGFR-Mutated Lung Cancer via Epithelial–Mesenchymal Transition

**DOI:** 10.3390/cancers15041092

**Published:** 2023-02-08

**Authors:** Kentaro Tokumo, Takeshi Masuda, Taku Nakashima, Masashi Namba, Kakuhiro Yamaguchi, Shinjiro Sakamoto, Yasushi Horimasu, Shintaro Miyamoto, Hiroshi Iwamoto, Kazunori Fujitaka, Yoshihiro Miyata, Morihito Okada, Hironobu Hamada, Noboru Hattori

**Affiliations:** 1Department of Molecular and Internal Medicine, Graduate School of Biomedical and Health Sciences, Hiroshima University, 1-2-3 Kasumi, Minami-ku, Hiroshima 734-8551, Japan; 2Department of Clinical Oncology, Hiroshima University Hospital, 1-2-3 Kasumi, Minami-ku, Hiroshima 734-8551, Japan; 3Department of Respiratory Medicine, Hiroshima University Hospital, 1-2-3 Kasumi, Minami-ku, Hiroshima 734-8551, Japan; 4Department of Surgical Oncology, Research Institute for Radiation Biology and Medicine, Hiroshima University, Hiroshima 734-8551, Japan; 5Department of Physical Analysis and Therapeutic Sciences, Hiroshima University, 1-2-3 Kasumi, Minami-ku, Hiroshima 734-8551, Japan

**Keywords:** epidermal growth factor receptor, plasminogen activator inhibitor-1, drug-tolerant cells, non-small-cell lung cancer, epithelial–mesenchymal transition

## Abstract

**Simple Summary:**

Osimertinib is widely employed in patients with epidermal growth factor receptor (EGFR)-mutated non-small-cell lung cancer (NSCLC). Most EGFR-mutated NSCLC cells are killed within a few days after osimertinib treatment; however, surviving cells remain detectable and are called drug-tolerant cells. Avoiding drug tolerance would maintain the long-term efficacy of osimertinib. We showed that the expression of PAI-1 and mesenchymal genes in EGFR-mutated cancer cell lines was upregulated after developing tolerance to EGFR-TKIs in vitro. In addition, PAI-1 inhibition limited the proliferation and mesenchymal gene expression of EGFR-TKI-tolerant cells. These results indicate that PAI-1 is involved in drug tolerance to EGFR-TKIs via epithelial–mesenchymal transition. Furthermore, we demonstrated that the combination of osimertinib with a PAI-1 inhibitor prevented the regrowth of osimertinib-treated tumors composed of EGFR-mutated cancer cells in in vivo experiments. Based on these observations, PAI-1 may prove to be a potential therapeutic target for overcoming tolerance to osimertinib.

**Abstract:**

Most epidermal growth factor receptor (EGFR)-mutated non-small-cell lung cancer (NSCLC) cells are killed within a few days after osimertinib treatment; however, surviving cells remain detectable and are called drug-tolerant cells. Plasminogen activator inhibitor-1 (PAI-1) was reported to be involved in chemotherapeutic or radiotherapeutic resistance. The purpose of the present study was to investigate whether PAI-1 is involved in osimertinib tolerance and whether it could be a therapeutic target for overcoming this tolerance. We showed that the PAI-1 mRNA expression levels and mesenchymal gene expression levels were significantly higher in drug-tolerant EGFR-mutated NSCLC cells than in control cells after 7 days of in vitro osimertinib treatment. Additionally, an RNA microarray analysis revealed upregulation of the integrin-induced EMT pathway in osimertinib-tolerant cells. Furthermore, we observed that PAI-1 inhibitors suppressed proliferation and the degree of epithelial–mesenchymal transition (EMT) in tolerant cells. Finally, in a subcutaneous tumor model, we showed that combining osimertinib with a PAI-1 inhibitor prevented the regrowth of tumors comprising EGFR-mutated cancer cells. The present study is the first to show PAI-1 to be involved in tolerance to osimertinib via EMT.

## 1. Introduction

Lung cancer is the most lethal cancer globally [1]. Its histopathological types are divided into small-cell lung cancer (SCLC) and non-small-cell lung cancer (NSCLC), with frequencies of 15–20% and 80–85%, respectively [2]. NSCLC is further subclassified into adenocarcinoma (50%), squamous cell carcinoma (30%), and large-cell carcinoma (5%). Approximately 40–50% and 10–15% of patients with lung adenocarcinoma in Asian and Western countries, respectively, harbor an active mutation in the gene encoding epidermal growth factor receptor (EGFR) [3]. Several EGFR tyrosine kinase inhibitors (TKIs) have been approved for the treatment of NSCLC. They include the first-generation gefitinib and erlotinib and the second-generation afatinib and dacomitinib. The response rate of these drugs in patients with EGFR-mutated NSCLC has been reported to be approximately 70% [4,5,6], and these drugs are the standard first-line treatment for EGFR-mutated NSCLC [7]. In contrast, most cases showed progressive disease with acquired resistance to EGFR-TKIs; approximately 50% of these cases manifested an EGFR exon 20 T790M mutation as the prevalent resistance mechanism [8]. To address this issue, a “third-generation” EGFR-TKI, osimertinib, which is a potent inhibitor of active EGFR mutation and T790M resistance mutations, has been approved for use. Based on the results of a clinical trial, osimertinib is recommended as the standard treatment for patients harboring a T790M mutation [9]. Furthermore, osimertinib is the preferred first-line treatment agent for EGFR-mutated NSCLC [7,10].

Although osimertinib is widely employed in patients with EGFR-mutated NSCLC, most patients exhibit progressive disease. Previous studies have demonstrated various mechanisms of acquired resistance to osimertinib, such as a second-site mutation of EGFR, e.g., a C797S mutation or EGFR-independent resistance mechanisms; amplification of HER2, MET, and KRAS; and bypass pathway activation of MAPK, ERK, and AKT signaling [8]. Additionally, previous studies have shown that EMT is associated with EGFR-TKI resistance, including the acquired resistance to osimertinib [11,12,13,14,15,16,17,18]. Separately, the concept of drug-tolerant cells (DTCs) has been considered important in osimertinib resistance [19]. DTCs are viable cells that are observed immediately after the initiation of anticancer therapy, which can acquire treatment resistance over time. In contrast, treatment-resistant cells are actively proliferating cells that are inherently non-responsive to a particular therapy or acquire resistance over time after anticancer treatment. Several prior studies have shown that most EGFR-mutated lung cancer cells die within a few days after EGFR-TKI treatment; however, DTCs remain detectable [20]. In addition, these studies have reported the underlying mechanism of drug tolerance, including the bypass signal of the EGFR pathway (IFG-1R, AXL, Notch, and FGFR3) [19]. Moreover, only one study has demonstrated the association between EMT and drug tolerance [21]. It is hypothesized that these EGFR-TKI DTCs are an important source of cancer cells that will acquire resistant mechanisms to EGFR-TKIs [22]. In this context, we focused on tolerant cells immediately after the start of treatment rather than on cells that acquired treatment resistance over time and suggest a novel treatment strategy for avoiding drug tolerance that could result in preventing acquired resistance to osimertinib. Subsequently, avoiding drug tolerance would maintain the long-term efficacy of osimertinib.

PAI-1 is a 47 kDa glycoprotein produced by vascular endothelial cells, adipocytes, and fibroblasts. PAI-1 physiologically hinders plasminogen activators and functions as an inhibitor of fibrinolysis. It is also reported to be involved in systemic inflammation, atherosclerosis, and fibrosis in several organs [23]. Conversely, several cancer cell types produce PAI-1, and numerous in vitro and in vivo studies have reported its association with cancer progression through direct pro-proliferative and anti-apoptotic effects [24,25]. Previous studies have reported its involvement in chemotherapeutic or radiotherapeutic resistance in cancers through EMT [26,27].

Based on these observations, we investigated the involvement of PAI-1 in tolerance to osimertinib via EMT in EGFR-mutated NSCLC and its likelihood of serving as a therapeutic target to overcome this tolerance.

## 2. Materials and Methods

### 2.1. Cell Lines

The human EGFR-mutated lung cancer cell line PC-9, with an exon 19 deletion mutation, was provided by RIKEN BRC through the National Bio-Resource Project of the MEXT/AMED, Japan. The human EGFR-mutated lung cancer cell line H1975, with an L858R/T790M mutation was purchased from the American Type Culture Collection (Manassas, VA, USA). These cell lines were obtained directly from the cell banks, and cell authentication and mycoplasma contamination exclusion were already checked. These cells were resuscitated immediately after acquisition, properly divided, and cryopreserved, and all cell experiments were performed within eight passages. We also used EGFR-mutated cells obtained from one patient’s pleural effusion with advanced EGFR exon 19 deletion positive lung adenocarcinoma. The patient manifested a T790M mutation 7 months after afatinib administration. Prior written informed consent was obtained from the patient for use of this patient-derived cell line. Procedures for the isolation and culture of cancer cells are described in the Appendix A.

### 2.2. Patients

We retrospectively reviewed the medical records of patients with unresectable lung adenocarcinoma and EGFR mutations treated at Hiroshima University Hospital between August 2002 and December 2020. The patients who received gefitinib, erlotinib, or afatinib and underwent rebiopsy after treatment failure with these drugs were enrolled. Patients who met the described criteria were enrolled, irrespective of their age and sex. For the immunohistochemical staining analysis of PAI-1 expression, we excluded patients whose tissue samples were obtained via bronchofiberscopy.

### 2.3. Cell Culture

The cells were cultured in RPMI-1640 medium supplemented with 10% fetal bovine serum, penicillin, and streptomycin (100 U/mL and 100 µg/mL, respectively) under a humidified 5% CO_2_ atmosphere at 37 °C in an incubator. Cancer cells were seeded at a density of 1 × 10^5^ cells/well in six-well plates for mutation, gene copy number, and microarray analyses. The cells were used for various analyses 36 h after incubation.

### 2.4. Reagents

The PAI-1 inhibitor SK-216 (Appendix A) was chemically synthesized and supplied by Shizuoka Coffein Co., Ltd. (Shizuoka, Japan). The IC_50_ for SK-216 cells was determined to be 44 μM, as reported in the international patent, WO04/010996. Two EGFR-TKIs, gefitinib (Selleck Chemicals, Houston, TX, USA) and osimertinib (Selleck Chemicals), were used.

### 2.5. Quantitative RealTime PCR (qRT-PCR)

Total cellular RNA was isolated using an RNeasy Mini Kit (Qiagen, Venlo, The Netherlands) according to the manufacturer’s instructions. The isolated total RNA was reverse-transcribed into cDNA using a High Capacity RNA-to-cDNA Kit (Thermo Fisher Scientific, Waltham, MA, USA). qRT-PCR was performed using CFX384 (Bio-Rad, Hercules, CA, USA). mRNA expression levels were evaluated and normalized to β-actin levels as an endogenous reference. All primers for PCR were obtained from the TaqMan Gene Expression Assays (Thermo Fisher Scientific). The primers that were used are described in the Appendix A.

### 2.6. Quantification of PAI-1 Protein

The PAI-1 protein in the culture medium was evaluated by an enzyme-linked immunosorbent assay (ELISA), and the intracellular level of PAI-1 protein was estimated using Western blotting. ELISA: The cancer cells were seeded at a density of 1 × 10^5^ cells/well in six-well plates. The total PAI-1 secreted into serum-free culture medium for 36 h was measured using an ELISA kit (R&D Systems, Minneapolis, MN, USA) according to the manufacturer’s instructions. To correct for differences in cell number, the measured concentration was divided by the number of cells (per 10^6^ cells). Western blotting was performed as follows: Total cell lysates were prepared using lysis buffer (150 mM NaCl, 1% NP-40, and 50 mM Tris-HCl) supplemented with a 1% protease inhibitor cocktail (Merck, Darmstadt, Germany). The lysates were centrifuged at 12,000 rpm for 10 min at 4 °C. The protein concentration was determined using the supernatant and the BCA protein assay kit (Thermo Fisher Scientific). Protein extracts were loaded on 4–15% gradient SDS-PAGE gels (Mini-PROTEAN TGX Precast Gels; Bio-Rad) at 10 µg of protein per lane. After electrophoresis, proteins were electro-transferred onto a polyvinylidene difluoride membrane (Immobilon-P transfer membrane, Merck). The membrane was blocked with Blocking One (Nacalai tesque, Kyoto, Japan) for 1 h and incubated overnight with an anti-PAI-1 Rabbit mAb (#11907, Cell Signaling Technology, Danvers, MA, USA) and an anti-GAPDH Rabbit mAb (#5174, Cell Signaling Technology) conjugated with Can Get Signal Solution 1 (Toyobo, Osaka, Japan). The membrane was then washed and incubated with a secondary antibody conjugated with Can Get Signal Solution 2 (Toyobo). After further washing, the membrane was incubated with ImmunoStar Zeta (FUJIFILM Wako Chemical, Tokyo, Japan) and exposed using a WSE-6100 LuminoGraph I imaging system (ATTO, Tokyo, Japan). The band intensities were analyzed densitometrically using ImageJ (National Institutes of Health, Bethesda, MD, USA).

### 2.7. Immunohistochemical Staining

We performed immunohistochemical staining according to a previous study [26]. The stained specimens were checked, and micrographs were obtained under the same photographic conditions at three representative locations where the aggregation of cancer cells could be clearly seen in each specimen and where the DAB coloration was considered the strongest. Micrographs were obtained using a BZ-9000 microscope (Keyence, Osaka, Japan). The open-source plugin IHC Profiler [28], which is compatible with ImageJ, was used to separate DAB- and hematoxylin-stained regions. Then, the percentage of regions stained with DAB was calculated and compared.

### 2.8. Generation of EGFR-TKI-Tolerant Cells

The cells (1 × 10^5^ cells/well) were plated in six-well plates and incubated for 24 h. The medium was then replaced with a medium containing 1 µM gefitinib or 0.5 µM osimertinib with 30–50 times the established IC_50_ values. Starting from this time point as day 1, the medium was changed every 3 days (day 4 and day 7), and on day 8, the surviving cells were determined to be EGFR-TKI-tolerant cells [29].

### 2.9. EGFR T790M, C797S Mutation, and Gene Copy Number Analyses

Total DNA from EGFR-TKI-tolerant cells was isolated using a DNeasy Blood & Tissue kit (Qiagen) according to the manufacturer’s instructions. Genomic samples were sent to commercial laboratories (SRL, Inc., Tokyo, Japan) for mutation testing of the EGFR T790M and C797S mutations (peptide nucleic-acid-locked nucleic acid clamp PCR or the Cobas method). The *MET* and *HER2* gene copy numbers in the total DNA were assessed via qRT-PCR using TaqMan copy number assays (Thermo Fisher Scientific). Quantitative RT-PCR was performed using a 7900HT Fast Real-Time PCR system (Thermo Fisher Scientific). The primers that were used are described in the Appendix A.

### 2.10. Validation of PAI-1-siRNA Knockdown Efficacy

H1975 or PC-9 cells were transfected at 60% confluence with 10 nM Silencer Select small interfering RNA (siRNA) targeting human PAI-1 (PAI-1-siRNA, siRNA ID: s10013 and s10014) or a Silencer Select negative control siRNA (NC-siRNA) (Thermo Fisher Scientific). The Lipofectamine RNAiMAX transfection reagent (Thermo Fisher Scientific) was used for transfection. To eliminate the effect of PAI-1 accumulated in the culture medium, the cells were supplied with a serum-free culture medium prior to the knockdown operation.

### 2.11. Proliferation Assay

We used SK-216 to inhibit PAI-1. The EGFR-TKI-tolerant cells were seeded in 24-well plates at 3 × 10^4^ cells/well. After 24 h, they were replaced with a culture medium supplemented with gefitinib (1 µM) or osimertinib (0.5 µM) and SK-216 (50 or 100 μM), and the number of viable cells in each well was counted every 24 h using trypan blue. An experimental method using siRNA is described below. After siRNA exposure, the number of viable cells in each well was counted every 24 h using trypan blue. To eliminate the effect of PAI-1 protein in the culture medium, the culture medium was replaced with a serum-free medium immediately before siRNA administration.

### 2.12. Analysis of PAI-1- or EMT-Related Gene Expression

H1975 or PC-9 cells (4 × 10^5^) were plated in six-well plates and incubated for 24 h. The medium was then replaced with a medium containing or lacking 0.5 µM osimertinib and 0.5 µM osimertinib with SK-216 (50 or 100 µM). The media were changed every 3 days until day 8. The cells were collected, and the mRNA expression levels were compared using RT-PCR.

### 2.13. Microarray

Total RNA was extracted using an RNeasy Mini Kit (Qiagen) according to the manufacturer’s instructions and quantified using an Agilent 2100 bioanalyzer (Agilent). The global expression of mRNAs in the cells was examined using a Clariom S Array (Thermo Fisher Scientific). The raw mRNA expression profile data were analyzed using iDEP version 0.93 [30]. A parametric analysis of gene set enrichment (PAGE) was used to identify genes that were differentially expressed between the two groups. The Kyoto Encyclopedia of Genes and Genomes (KEGG) pathway database and diagrams were used to visualize and compare the expression of pathways [31]. Genes with adjusted *p* values < 0.05 after false discovery rate correction and a fold change of >2 or <0.5 were considered to be differentially expressed.

### 2.14. Animal Experimental Procedure

NOD.CB17-Prkdcscid /J (NOD SCID) mice (6 weeks old, 17–20 g, female) were used (Charles River Laboratories Japan, Yokohama, Japan). First, 1 × 10^6^ H1975 cells were suspended in 80 μL of RPMI-1640 medium mixed with 160 μL of Matrigel Basement Membrane Matrix (Corning, Corning, NY, USA) and subcutaneously injected into the backs of the NOD SCID mice. After 10 days of tumor incubation, the mice were randomly divided into four groups: a vehicle group, an SK-216 treatment group, an osimertinib treatment group, and an osimertinib + SK-216 treatment group. Methylcellulose (0.5%) was used as a vehicle control. The average body weights of the mice in each group were not different at the beginning of the intervention. Osimertinib (5 mg/kg) was prepared using 0.5% methylcellulose. The vehicle control or osimertinib were administered orally by gavage 6 days per week. The SK-216 and osimertinib + SK-216 groups were provided drinking water containing SK-216 (100 ppm = 187.4µM). These administrations were performed in an open-label setting. The tumor volume was calculated using the empirical formula V = 1/2 × ([shortest diameter]^2^ × [the longest diameter]). After 13 days of observation following the start of the intervention, subcutaneous tumors were collected, and the expression of PAI-1 was evaluated using the immunostaining method described above. To investigate the efficacy of SK-216 in inhibiting the growth of tumors resistant to osimertinib after long-term treatment, the same intervention was performed under the same conditions and observations were made for 11 weeks.

### 2.15. Statistical Analysis

All statistical analyses were performed using JMP Pro 15 (SAS Institute, Cary, NC, USA). All results are expressed as means ± standard errors (SEs) of the means. Student’s *t*-test was used for two-group comparisons. A one-way ANOVA with Tukey’s test was used to evaluate statistical differences for comparisons with more than three groups. Statistical significance was set at *p* < 0.05.

## 3. Results

### 3.1. The Level of PAI-1 Increased in EGFR-Mutated Cancer Cells with Tolerance to EGFR-TKI

To confirm the increase in PAI-1 expression in cancer cells with tolerance to EGFR-TKIs, we compared the PAI-1 expression in EGFR-TKI-tolerant cells to that in control cells in vitro. We showed that the PAI-1 mRNA expression in osimertinib-tolerant H1975 cells with L858R/T790M mutations was significantly higher in EGFR-TKI-tolerant cells than in control cells (Figure 1A). Furthermore, the PAI-1 mRNA expression in gefitinib- or osimertinib-tolerant PC-9 cells with exon 19 deletion mutations was significantly higher than that in control cells (Figure 1B). We also observed that PAI-1 protein expression was significantly higher in EGFR-TKI-tolerant cells than in control cells (Figure 1C–G). These observations showed that the PAI-1 expression in EGFR-mutated cells increases when cancer cells develop tolerance to EGFR-TKIs. Both the H1975 and PC-9 resistant cells manifested thicker cell margins and slightly rounded shapes in comparison to the original cells (Figure 1H). Subsequently, the same investigation was conducted using cancer cells derived from one patient with an EGFR exon 19 deletion and T790M mutations. These cells had the T790M mutation, which is resistant to gefitinib. Thus, most of these cells survived after treatment with gefitinib, tolerance was not induced, and PAI-1 expression was not increased. In contrast, since osimertinib has antitumor efficacy on T790M-positive cells, most cells treated with osimertinib died, and only osimertinib-tolerant cells survived. Osimertinib-tolerant cells showed significantly increased PAI-1 mRNA expression compared with the corresponding mRNA expression in control cells (Figure 1I).

### 3.2. Involvement of PAI-1 in Tolerance to EGFR-TKI in EGFR-Mutated Lung Cancer Cells

To examine the involvement of PAI-1 in the proliferation of EGFR-mutated cancer cells with tolerance to EGFR-TKIs, we investigated the effect of PAI-1 inhibition on the proliferation of EGFR-TKI-tolerant cells. First, we investigated the efficacy of the PAI-1 inhibitor SK-216 with EGFR-TKI-treatment-naive cancer cells. We found that PAI-1 inhibition decreased the proliferation of PC-9 cells but not that of H1975 cells (Figure 2A,B). Subsequently, we revealed that PAI-1 inhibition significantly limited the proliferation of gefitinib- and osimertinib-tolerant H1975 and PC-9 cells (Figure 2C–E). We showed that the expression level of PAI-1 protein in the culture medium was not significantly different in osimertinib-tolerant cells with and without SK-216 treatment (Appendix A). This result aligned with our hypothesis, as SK-216 suppresses the function but not the expression of PAI-1 in cancer cells. Further, to investigate the direct effect of PAI-1 on tolerance to EGFR-TKI, we inhibited PAI-1 expression in these cells using siRNA. The levels of PAI-1 mRNA expression in the tolerant cells were significantly decreased in PAI-1 knockdown cells compared to control cells (Appendix A). We observed that PAI-1 knockdown significantly decreased the proliferation of the tolerant PC-9 and H1975 cells compared to control cells (Figure 2F–I). These observations showed the possible involvement of PAI-1 in tolerance to EGFR-TKIs. Simultaneously, we examined whether cells tolerant to EGFR-TKIs acquired additional genetic mutations or amplifications, such as MET amplification, HER2 amplification, the EGFR T790M mutation, and the C797S mutation. We found that PC-9 and H1975 cells, after EGFR-TKI treatment, showed no significant differences in the gene copy numbers of MET and HER2 compared with the corresponding gene copy numbers seen in the control cells (Appendix A). In addition, PC-9 cells were not found to harbor any T790M or C797S mutations after the gefitinib or osimertinib treatments. In H1975 cells harboring the T790M mutation, C797S was not observed after the osimertinib treatment.

### 3.3. PAI-1 Was Found to Be Involved in Tolerance to Osimertinib in EGFR-Mutated Lung Cancer via Its Association with EMT

We investigated the involvement of PAI-1 in tolerance to EGFR-TKIs via its association with EMT. In the present study, we examined the association between PAI-1 and osimertinib-induced tolerance because osimertinib is the standard drug for advanced EGFR-mutated NSCLC. We administered osimertinib to H1975 and PC-9 cells and evaluated the mRNA expression of EMT-related genes. The mRNA expression of mesenchymal markers among EMT-related genes and PAI-1 was significantly upregulated after osimertinib treatment in these cells (Figure 3A,B). Conversely, SK-216 decreased the mRNA expression of these mesenchymal marker genes in H1975 cells (Figure 3C). A partial decrease in mesenchymal gene expression was observed in the PC-9 cells (Figure 3D). These findings suggested that PAI-1 is involved in osimertinib tolerance via its association with EMT and that the degree of this association depends on the cell type.

### 3.4. PAI-1 Was Found to Be Involved in Tolerance to Osimertinib via Its Association with Integrin-Initiated EMT

To investigate the mechanism through which PAI-1 is associated with EMT, we performed a comprehensive comparative genetic analysis between osimertinib-tolerant and control PC-9 or H1975 cells using microarrays. A pathway analysis was performed with PAGE using the KEGG pathway datasets. We showed a partial upregulation of the TGF-β-induced EMT pathway in EGFR-TKI-tolerant cells (Appendix A). Conversely, the pathway analysis revealed that the expression of genes related to the “regulation of actin cytoskeleton” and “focal adhesion” pathways was upregulated (Appendix A). A detailed review of the microarray results revealed that the expression of integrins αVβ3, αVβ6, and αVβ8 was upregulated in osimertinib-tolerant cells, accompanied by increased expression of some extracellular matrix components such as fibronectin and vitronectin, which bind to these integrins (Figure 4A,B). In addition, these genes included numerous members of the integrin-initiated EMT signaling pathway. These findings were confirmed via RT-PCR (Figure 4C,D). Subsequently, to investigate whether PAI-1 is involved in the integrin-initiated EMT signaling pathway, we compared the mRNA expression of genes in this pathway between the PAI-1-inhibited osimertinib-tolerant H1975 cell group and the control group using microarray analysis. The results of the microarray analysis suggested an increased expression of integrins and some extracellular matrix components in non-PAI-1-inhibited osimertinib-tolerant H1975 cells. On the contrary, PAI-1-inhibited osimertinib-tolerant cells showed a decrease in the expression of integrins and extracellular matrix components (Figure 4E). Some of the PAI-1-inhibition-induced decreased expression patterns observed in H1975 cells were also observed in PC-9 cells (Figure 4F). In addition, we showed that the mRNA expression of the integrin-initiated EMT signaling pathway was significantly downregulated in PAI-1-inhibited osimertinib-tolerant H1975 cells compared with the corresponding mRNA expression in the control cells (Figure 4G). In PC-9 cells, these results were also partially observed (Figure 4H). Taken together, these results indicated that PAI-1 is involved in drug tolerance to EGFR-TKIs in EGFR-mutated cells via its association with integrin-initiated EMT. The degree of this association depended on the cell type.

### 3.5. PAI-1 Inhibition Limited Regrowth of Osimertinib-Treated Tumors in the Subcutaneous Tumor Model

To confirm the increase in PAI-1 expression in EGFR-TKI-tolerant cells in vivo, we established xenograft models with H1975 cells in which EMT signaling was strongly suppressed by PAI-1 inhibition in vitro. We compared PAI-1 expression between osimertinib-treated and control subcutaneous tumor specimens. After 10 days of treatment, PAI-1 expression was significantly higher in the drug-tolerant cells than in the control cells (Figure 5A–C). Subsequently, to investigate whether SK-216 treatment inhibits tumor regrowth by inhibiting the proliferation of cells tolerant to EGFR-TKI, we compared tumor growth over a prolonged period between four groups: an osimertinib-treated group, an SK-216-treated group, an osimertinib + SK-216-treated group, and a vehicle-treated group. Tumor growth was significantly suppressed in the osimertinib-treated group and the osimertinib + SK-216-treated group compared with the tumor growth rate in the control group (Figure 5D). In the osimertinib-treated group, tumor regrowth was seen at approximately day 30. Conversely, the tumor volumes in the osimertinib + SK-216-treated group did not increase. Similar to the cell culture experiments, we showed that lung cancer cells that were tolerant to osimertinib expressed high levels of PAI-1 (Figure 5B,C), indicating that PAI-1 inhibitors could suppress PAI-1-related EMT and inhibit the regrowth of osimertinib-treated tumors. In addition, we showed that PAI-1 expression significantly increased in osimertinib-acquired resistant tumors compared with the PAI-1 expression level in the control tumors (Figure 5E,F). However, PAI-1 expression was not different between the osimertinib and osimertinib + SK-216 treatment groups. Since SK-216 suppresses the function but not the expression of PAI-1, this result seems to be reasonable.

### 3.6. PAI-1 Expression in Tumor Specimens at the Time of Acquired Resistance to EGFR-TKIs

It was difficult to obtain clinical specimens containing drug-tolerant cells from small tumors during the response to EGFR-TKI. Therefore, we investigated the clinical involvement of PAI-1 in acquired resistance to EGFR-TKI in patients with EGFR-mutated NSCLC. To investigate whether PAI-1 expression in the tumor tissue increased at the time of acquired resistance to EGFR-TKI, we compared the PAI-1 expression in the tumor specimens before and after acquired resistance to EGFR-TKIs in in patients with EGFR-mutated NSCLC. We enrolled 23 patients who had biopsies performed both at diagnosis and at the time the disease became resistant to EGFR-TKI. Eight patients were excluded since their tumor samples were not large enough to perform immunohistochemical staining. Subsequently, ten more patients were excluded since their specimens were very small samples obtained via bronchofiberscopy. Finally, we were able to investigate the PAI-1 expression in tumor specimens from five patients (Figure 6A,B and Appendix A). The clinical characteristics of the five patients are listed in Table 1. As shown in Figure 6C, the degree of PAI-1 expression was significantly higher in the tumor after acquired resistance to EGFR-TKI than during pretreatment (PAI-1-positive areas of 2.5% vs. 24.1%, respectively, *p* < 0.05). This finding, as well as the results of the in vivo experiments, suggested that PAI-1 involves both EGFR-TKI tolerance and acquired resistance. While the results obtained here are interesting, it is desirable to investigate the immunohistochemical analysis of PAI-1 using more samples. In addition, in the future, we would aim to clarify the associations between some genetic mutations and the effect of PAI-1 on EGFR-TKI tolerance.

## 4. Discussion

We showed that the expression of PAI-1 and mesenchymal genes in EGFR-mutated cancer cell lines was upregulated after developing tolerance to EGFR-TKIs in vitro. In addition, PAI-1 inhibition limited the proliferation and mesenchymal gene expression of EGFR-TKI-tolerant cells. These results indicate that PAI-1 is involved in drug tolerance to EGFR-TKIs via EMT (Figure 7A). Furthermore, we demonstrated that the combination of osimertinib with a PAI-1 inhibitor prevented the regrowth of osimertinib-treated tumors composed of EGFR-mutated cancer cells in in vivo experiments. Based on these observations, PAI-1 may prove to be a potential therapeutic target for overcoming tolerance to osimertinib and improving the prognosis of patients with EGFR-mutated NSCLC.

We showed that PAI-1 is involved in tolerance to osimertinib in EGFR-mutated cancer cells. It has been reported that PAI-1 is involved in tumor progression in patients with lung cancer, pleural malignant mesothelioma, breast cancer, esophageal cancer, gastric cancer, and renal cancer [32,33,34,35,36,37,38]. In addition, previous studies have reported the involvement of PAI-1 in resistance to cytotoxic chemotherapy and radiotherapy [27,39]. Therefore, to the best of our knowledge, the present study is the first to demonstrate the role of PAI-1 in tolerance to osimertinib in EGFR-mutated NSCLC. The median progression-free survival of patients with treatment-naive EGFR-mutated NSCLC and patients who acquired resistance to first- or second-generation EGFR-TKIs who received osimertinib was 19 or 10 months, respectively [9,10]. Only a few patients achieve complete remission despite being administered the standard treatment regimen for EGFR-mutated NSCLC. Our study’s findings indicate that PAI-1 might serve as a novel therapeutic target for overcoming tolerance to osimertinib and improving the prognosis of patients with EGFR-mutated NSCLC.

The results of our study indicated that PAI-1 is involved in osimertinib tolerance via EMT in EGFR-mutated cancer cells. Previously, we demonstrated that PAI-1 is involved in EMT in NSCLC and pulmonary alveolar cells [26,40]. In addition, several prior studies have shown EMT to be associated with chemotherapeutic and radiotherapeutic resistance and have shown the involvement of PAI-1 in the EMT process [27,39]. A previous study showed the association of EMT with osimertinib tolerance [21]. Consequently, we investigated whether PAI-1 is involved in EMT during osimertinib tolerance and found that PAI inhibition suppressed the mRNA expression of mesenchymal marker genes in osimertinib-tolerant cells. The degree of this involvement depended on the types of cells. Separately, as shown in Figure 3, E-cadherin was not upregulated after SK-216 administration, which may not be consistent with the process of the typical EMT. Recently, the concept of partial EMT, which indicates a transitional or intermediate state between epithelial and mesenchymal properties, has been proposed [40]. We consider that the osimertinib-tolerant cells showed a state of partial EMT, during which only mesenchymal markers changed rapidly and epithelial markers did not respond well.

The present study is novel in elucidating that PAI-1 is involved in EMT through its association with integrin and not through the TGF pathway (Figure 7B). We previously showed PAI-1 to be involved in the EMT of the NSCLC cell line A549 and the alveolar epithelial cell line LA-4 [26,41,42]. In the EMT processes, involvement of TGF-β-induced EMT pathways, such as Snail and ZEB1, was not observed [42]. Furthermore, the results of the present study showed that TGF-β-induced EMT signaling pathways were only partially upregulated. In contrast, the integrin-dependent EMT signaling pathway was upregulated. Integrins have been reported to be involved in the EMT of cancer cells, resulting in the promotion of metastasis and invasion [43,44,45]. In addition, a previous study showed that integrin-dependent EMT induces EGFR-TKI resistance [46]. These studies support the results of our study, which showed that PAI-1 induces osimertinib tolerance through integrin-dependent EMT. In contrast, numerous previous studies have reported that the TGF-β-induced EMT signaling pathway is associated with osimertinib resistance [11,12,13,14,15,16,17,18]. This discrepancy could be attributed to the exposure times of cancer cells to osimertinib until the analysis of the EMT pathway. The exposure times in our study and in previous studies were 8 days or several weeks or months. We consider that PAI-1-related integrin-dependent EMT plays an important role in the tolerance to osimertinib.

PAI-1 appears to induce EMT signaling by upregulating the expression of integrins. The results of our microarray analysis showed that osimertinib-tolerant cells expressed extracellular matrix (ECM) components such as vitronectin, fibronectin, laminin, and collagen. In addition, PAI-1, which is expressed by osimertinib-tolerant cells, can protect the ECM from proteolysis by inhibiting the local plasminogen activator, resulting in the accumulation of ECM [47]. Subsequently, ECM-to-cell adhesion results in the formation of focal adhesions, which contain integrin and increase the expression of integrin [48]. As described above, we propose that PAI-1 induces the accumulation of ECM, and the expression of integrin is enhanced, eventually leading to osimertinib tolerance.

However, the detailed molecular mechanism requires further study.

In this study, the EGFR-TKI treatment induced high PAI-1 expression in EGFR-mutated cancer cells. PAI-1 gene expression is regulated by a wide variety of cytokines and growth factors, including TGF-β, EGF, interleukin-1β, and insulin [49]. In addition, EGFR-TKI treatment is reported to induce reactive oxygen species (ROS) in cancer cells, which in turn, can induce PAI-1 expression [50,51]. In this study, we showed, using a microarray analysis, that expression levels of TGF-β, EGF, interleukin-1β, and insulin were not clearly increased in the osimertinib-tolerant cells compared to the control cells. Therefore, osimertinib-induced ROS in cancer cells may be involved in the expression of PAI-1 in cancer cells. Further studies are needed to clarify the mechanism between EGFR-TKI treatment and PAI-1 expression variations.

Since we have shown that PAI-1 is involved in EGFR-TKI tolerance, serum PAI-1 levels might be a useful predictor of resistance to EGFR-TKI and the therapeutic response to EGFR-TKI. According to our review, one previous study reported that the serum PAI-1 level prior to EGFR-TKI treatment could be a surrogate marker for the therapeutic effect of EGFR-TKI [52]. This research shows that EGFR-mutation-positive cells express PAI-1 through the Notch3-dependent β-catenin signaling induced by EGFR-TKI. In cases with lower pretreatment serum PAI-1 levels, the serum PAI-1 levels at the time of resistance tended to increase. In addition, this study showed that lower pretreatment serum PAI-1 levels were associated with a poorer response to EGFR-TKI. However, prospective studies with a larger number of patients are needed to determine whether serum PAI-1 levels can act as a surrogate marker of EGFR-TKI treatment efficacy.

## 5. Conclusions

The present study is the first to show PAI-1 to be involved in tolerance to osimertinib via EMT. Furthermore, PAI-1 may serve as a potential therapeutic target for overcoming osimertinib tolerance and improving the prognosis of patients with EGFR-mutated NSCLC.

## Figures and Tables

**Figure 1 cancers-15-01092-f001:**
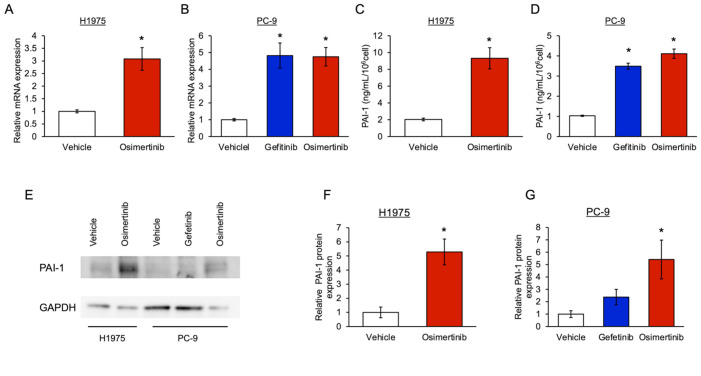
PAI-1 expression in EGFR-TKI-tolerant cancer cells. (**A**,**B**) qRT-PCR analysis of the expression of PAI-1 mRNA in EGFR-TKI-tolerant PC-9 and H1975 cells compared with the corresponding mRNA expression in the vehicle controls. Relative expression level of PAI-1 is shown with standard error bars (n = 3); * *p* < 0.05 compared with cells exposed to vehicle controls (Student’s *t*-test or one-way ANOVA with Tukey’s test). (**C**,**D**) Concentration of PAI-1 protein in the culture medium of EGFR-TKI-tolerant PC-9 and H1975 cells compared with the PAI-1 protein concentration seen in the controls. Mean concentrations are shown with standard error bars (n = 5); * *p* < 0.05 compared with cells exposed to vehicle controls (Student’s t-test or one-way ANOVA with Tukey’s test). (**E**) Western blotting of PAI-1 in EGFR-TKI-tolerant PC-9 and H1975 cells compared with the vehicle controls. (**F**,**G**) Levels of PAI-1 protein normalized to GAPDH in EGFR-TKI-tolerant PC-9 and H1975 cells compared with the vehicle controls. The relative expression level of PAI-1 is shown with standard error bars (n = 3); * *p* < 0.05 compared with cells exposed to the vehicle controls (Student’s t-test or one-way ANOVA with Tukey’s test). (**H**) Micrographs of normal and osimertinib-tolerant cells. Upper-left: normal H1975, upper-right: osimertinib-tolerant H1975 cells; lower-left: normal PC-9 cells; lower-right: osimertinib-tolerant PC-9 cells; bar: 40 µm. (**I**) qRT-PCR analysis of the expression of PAI-1 mRNA in EGFR-TKI-tolerant and control cancer cells derived from a patient with an EGFR exon 19 deletion and the T790M mutation seven months after afatinib administration. As in the other cell experiments, gefitinib and osimertinib were administered. The relative expression level of PAI-1 is shown with standard error bars (n = 3); * *p* < 0.05 compared with cells exposed to vehicle controls (one-way ANOVA with Tukey’s test). All the whole western blot figures can be found in the Appendix A.

**Figure 2 cancers-15-01092-f002:**
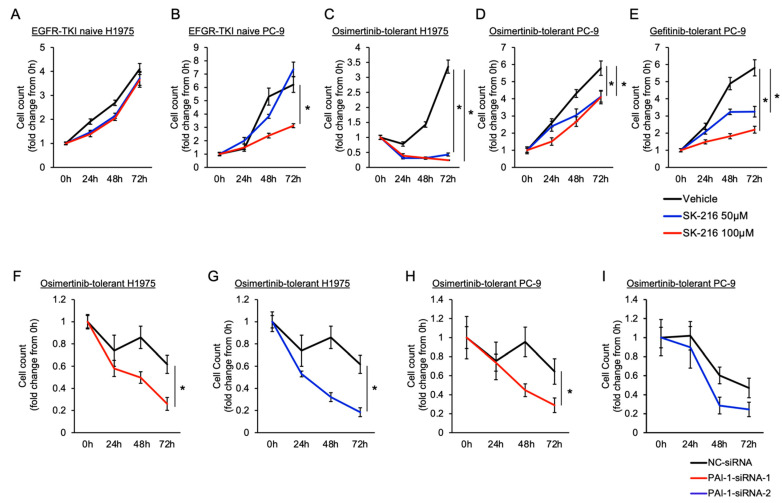
The effect of PAI-1 inhibition on the proliferation of EGFR-TKI-tolerant cells. The fold changes in the EGFR-TKI-naive (**A**) H1975 and (**B**) PC-9 cells between pretreatment and after treatment with 0, 50, or 100 μM SK-216, a PAI-1 inhibitor. The fold changes in (**C**) H1975 and (**D**,**E**) PC-9 cells with tolerance to gefitinib or osimertinib before and after PAI-1 inhibition. The black, blue, and red lines represent the changes observed after treatment with 0, 50, and 100 µM SK-216, respectively. The mean fold changes relative to the starting point (0 h) are shown with standard error bars (n = 6). At 72 h, the mean fold changes between the 0 μM and 50 or 100 μM treatment regimens were statistically compared; * *p* < 0.05 compared with cells exposed to vehicle controls (one-way ANOVA with Tukey’s test). (**F–I**) Fold changes in (**F**,**G**) H1975 and (**H**,**I**) PC-9 cells with tolerance to osimertinib over time after exposure to siRNAs (PAI-1-siRNA (PAI-1-siRNA-1 and PAI-1-siRNA-2) or negative control siRNA (NC-siRNA)). The data for PAI-1-siRNA-1 and PAI-1-siRNA-2 are shown for the two siRNA reagents described in the Materials and Methods section, s10013 and s10014, respectively. The black, red, and blue lines represent the changes observed after exposure to siRNA, respectively. The mean fold changes relative to the starting point (0 h) are shown with standard error bars (n = 5). At 72 h, the mean fold changes between the PAI-1-siRNA and NC-siRNA exposures were statistically compared; * *p* < 0.05 compared with cells exposed to NC-siRNA (Student’s *t*-test).

**Figure 3 cancers-15-01092-f003:**
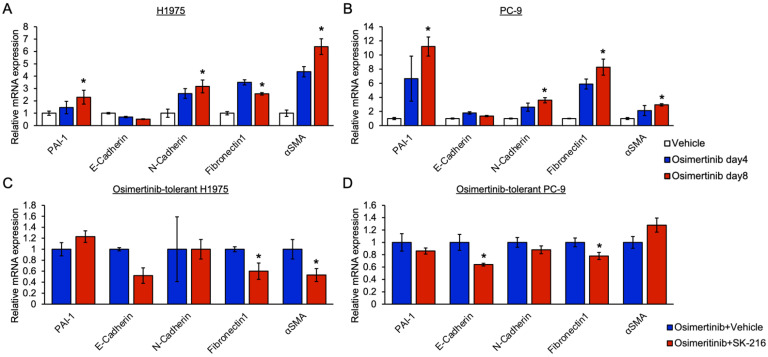
Involvement of PAI-1 in tolerance to osimertinib in EGFR-mutated lung cancer via its association with EMT. (**A**,**B**) qRT-PCR analysis of mRNA expression of PAI-1 and EMT-related genes in (**A**) H1975 and (**B**) PC-9 cells. The mRNA expression was measured on days 4 and 8 after osimertinib treatment. Relative expression level of each gene is shown with standard error bars (n = 4); * *p* < 0.05 compared with cells exposed to vehicle controls (one-way ANOVA with Tukey’s test). (**C**,**D**) qRT-PCR analysis of the mRNA expression of PAI-1 and EMT markers in osimertinib-tolerant (**C**) H1975 and (**D**) PC-9 cells. These osimertinib-tolerant cells were treated with SK-216 combined with osimertinib, and the mRNA expression of these genes was evaluated 48 h after SK-216 and osimertinib administration. The relative expression level of each gene is shown with standard error bars (n = 4); * *p* < 0.05 compared with cells exposed to vehicle controls (Student’s *t*-test).

**Figure 4 cancers-15-01092-f004:**
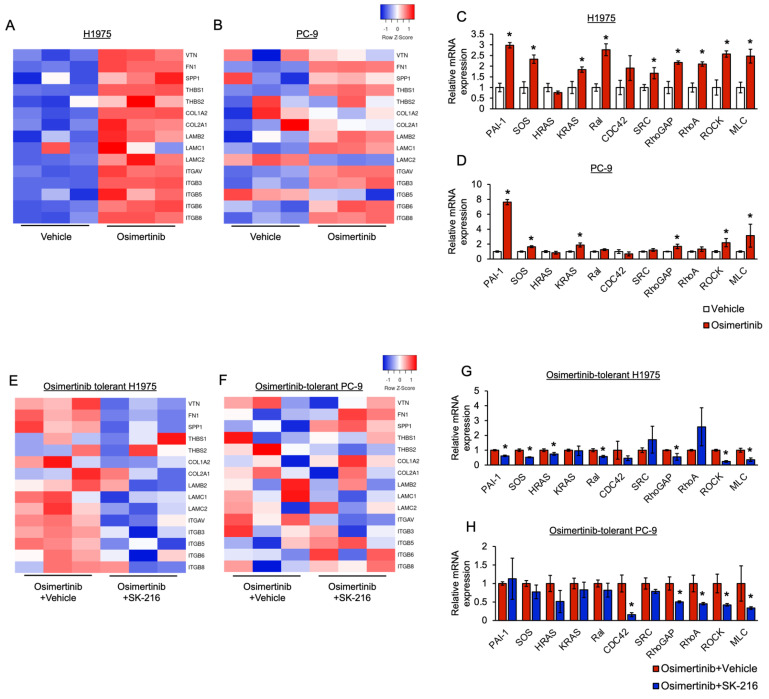
Involvement of PAI-1 in tolerance to EGFR-TKI via its association with integrin-initiated EMT. (**A**,**B**) From the microarray results, the mRNA expression levels of integrins αVβ3, αVβ6, and αVβ8 and extracellular matrix components were extracted and are shown in a heat map. In each of (**A**) H1975 and (**B**) PC-9, untreated controls were compared to osimertinib-tolerant cells. Blue represents downregulated genes; red represents upregulated genes. (**C**,**D**) qRT-PCR analysis of the mRNA expression of PAI-1 and integrin-initiated EMT-related genes in osimertinib-tolerant and untreated control (**C**) H1975 and (**D**) PC-9 cells. The relative expression level of each gene is shown with standard error bars (n = 4), compared with untreated control cells; * *p* < 0.05 (Student’s *t*-test). (**E**,**F**) From the microarray results, the mRNA expression levels of integrins and extracellular matrix components were extracted and are shown in a heat map. For (**E**) osimertinib-tolerant H1975 and (**F**) osimertinib-tolerant PC-9, untreated controls were compared to SK-216-treated cells. Blue represents downregulated genes; red represents upregulated genes. (**G**,**H**) qRT-PCR analysis of the expression of PAI-1 and integrin-initiated EMT-related genes in osimertinib-tolerant (**G**) H1975 and (**H**) PC-9 cells. Osimertinib-tolerant cells were treated with SK-216 combined with osimertinib, and mRNA expression was evaluated 48 h after SK-216 administration. The relative expression level of each gene is shown with standard error bars (n = 4); * *p* < 0.05 compared with cells exposed to vehicle controls (Student’s *t*-test).

**Figure 5 cancers-15-01092-f005:**
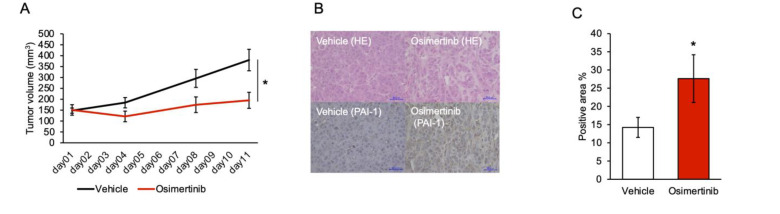
PAI-1 inhibition limited tolerance to osimertinib and regrowth in the subcutaneous tumor model. (**A**) H1975 cell tumor bearing mice were treated with vehicle (vehicle group) or osimertinib (osimertinib group), as described in the Methods Section (n = 6 each). Tumor volumes were measured over time from the start of treatment. Mean tumor volumes are shown with standard error bars; * *p* < 0.05 compared to the vehicle group (Student’s *t*-test). Representative images of tumor specimens stained with antibodies against human PAI-1 via immunohistochemistry. (**B**) The upper row is arranged with hematoxylin and eosin (HE) staining, and the lower row is arranged with PAI-1 immunostaining images. Tumors of the vehicle group and osimertinib group are shown; bar: 50 µm. (**C**) The proportion of PAI-1-positive areas between tumors of the vehicle group and osimertinib group. Data represent the mean values of six samples with standard error bars; * *p* < 0.05 compared to vehicle group (Student’s *t*-test). (**D**) H1975 cell tumor bearing mice were treated with vehicle (vehicle group), SK-216 (SK-216 group), osimertinib (osimertinib group), and osimertinib with SK-216 (osimertinib + SK-216 group), as described in the Methods Section (n = 8 each). A graph of the osimertinib group and osimertinib + SK-216 group is also shown in the upper right corner. Tumor volumes were measured over time from the start of treatment. Mean tumor volumes are shown with standard error bars; * *p* < 0.05 compared to the osimertinib group (one-way ANOVA with Tukey’s test). (**E**) The upper row is arranged with hematoxylin and eosin (HE) staining, and the lower row is arranged with PAI-1 immunostaining images. Tumors in the vehicle group, osimertinib group, and osimertinib + SK-216 group are shown; bar: 100 µm. (**F**) The proportion of PAI-1-positive area between tumors in the vehicle group and osimertinib group. Data represent the mean values of eight samples with standard error bars; * *p* < 0.05 compared with the vehicle group (Student’s *t*-test).

**Figure 6 cancers-15-01092-f006:**
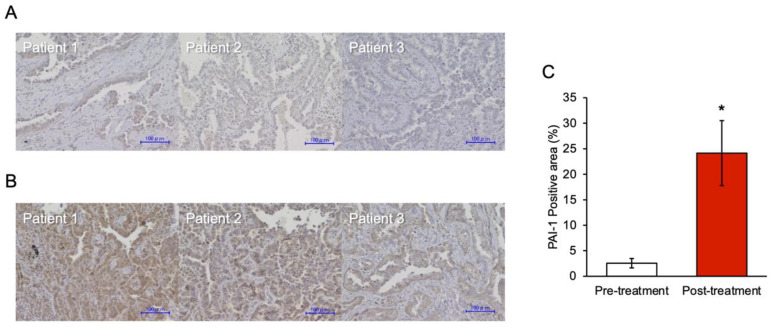
PAI-1 expression in the tumor specimens between pretreatment and time of acquired resistance to EGFR-TKIs. Representative images of tumor specimens stained with an antibody against human PAI-1 via immunohistochemistry: (**A**) at the time before the start of treatment (pretreatment) and (**B**) at the time of acquired resistance to EGFR-TKI (post-treatment). Magnification: ×200. Bar: 100 µm. (**C**) The proportion of PAI-1 positivity is the area between pretreatment and post-treatment. Data represent the mean values of five samples with standard error bars; * *p* < 0.05 compared with pretreatment (Student’s *t*-test).

**Figure 7 cancers-15-01092-f007:**
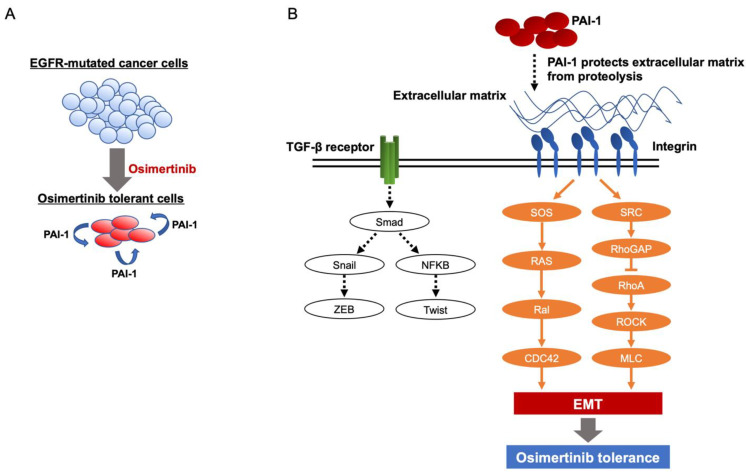
Schematic illustrations of our study. (**A**) The expression of PAI-1 in EGFR-mutated cancer cells increases after tolerance to osimertinib. PAI-1 is involved in tolerance to osimertinib in EGFR-mutated cancer cells. (**B**) PAI-1, which is expressed by osimertinib-tolerant cells, can protect the deposited extracellular matrix (ECM) from proteolysis, resulting in the accumulation of ECM. Subsequently, ECM-to-cell adhesion results in the formation of focal adhesions, which contain integrins. PAI-1 promotes tolerance to osimertinib via integrin- but not TGF-β-mediated EMT signaling pathways.

**Table 1 cancers-15-01092-t001:** Clinical characteristics of the five examined patients.

	Patient 1	Patient 2	Patient 3	Patient 4	Patient 5
Sex	F	M	M	M	M
Age (years)	66	65	52	66	63
Histological classification	ad	ad	ad	ad	ad
Clinical stage at diagnosis	IIIA	IIA	IIB	I	IIIB
EGFR mutation category	Exon 19 del	Exon 19 del	Exon 19 del	Exon 21 L858R	Exon 18 D719C
Tissue collection site (at diagnosis)	Lung	Lung	Lung	Lung	Lung
Method of tissue collection (at diagnosis)	Surgery	Surgery	Surgery	Surgery	Surgery
Tissue collection site (second biopsy)	Lung	Lung	Lung	Bone	Lymph node
Method of tissue collection (second biopsy)	Surgery	Surgery	Surgery	CT	Surgery
Presence of T790M mutation (second biopsy)	Yes	Yes	Yes	No	No
EGFR-TKI used before the second biopsy	Gefitinib	Gefitinib	Erlotinib	Afatinib	Afatinib
EGFR-TKI used after the second biopsy	Osimertinib	Osimertinib	Osimertinib	None	None

F: female, M: male, ad: adenocarcinoma, EGFR: epidermal growth factor receptor, del: deletion, CT: CT-guided biopsy, TKI: tyrosine kinase inhibitor.

## Data Availability

The data generated in the present study are available in this article and the Appendix A. The microarray data generated in the present study are publicly available in the Gene Expression Omnibus (GEO) at GSE200893 and GSE200894.

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
