# Peer review of "Association between Plasminogen Activator Inhibitor-1 and Osimertinib Tolerance in EGFR-Mutated Lung Cancer via Epithelial–Mesenchymal Transition"

_cancers, 2023, doi:10.3390/cancers15041092_

Round 1
Reviewer 1 Report (Previous Reviewer 1)
The authors focused on the role of plasminogen activator inhibitor-1 (PAI-1) in osimertinib tolerance in EGFR-mutated lung cancer via epithelial mesenchymal transition (EMT).
They also showed that combining osimertinib with a PAI-1 inhibitor prevented regrowth of tumors comprising EGFR-mutated cancer cells in vivo.
In the revised version of the manuscript, the author added the results as bellows;
Knockdown effect of PAI-1 on the proliferation of osimertinib-tolerant cells;
Function of SK-216, a PAI-1 inhibitor, functional inhibitor of PAI-1 rather than suppressing expression;
Microarray analysis results, osimertinib-tolerant cells vs those with PAI-1 inhibor
PAI-1 immunostaining results in vivo
PAI-1 expression of tumor specimens and PAI-1 concentrations in plasma samples from the patients before and after EGFR-TKI treatment
Osimertinib is currently the preferred first-line therapy in patients with NSCLC with common EGFR mutation. As a resistance to the treatment inevitably develops among patients treated with Osimertinib, understanding the mechanisms of resistance and the possible therapeutic options available is crucial to develop treatment applications for EGFR-mutated NSCLC resistant to osimertinib. While a hot topic in treatment for NSCLC patients is discussed in this manuscript, and the main research findings of this paper will be important, the reviewer has still annotated the manuscript with several concerns.
1. Arasada et al. reported that serum PAI-1 expression is low before EGFR-TKI, and high after treatment was evaluated (DOI: 10.1038/s41467-018-05626-2). They also described that patients expressing low levels of pre-treatment PAI-1 showed an increase in PAI-1 levels after EGFR TKI therapy and have a decreased progression free-survival. Please cite this paper and discuss this issue.
2. The label of Figure 1E is incorrect; osimertinib-tolerant PC-9 cells. In figure 1F, “gefitinib” means “samples from patients who were resistant to or after treatment of gefitinib?
3. In Figure 3, the mRNA expression of EMT markers as well as PAI-1 was significantly upregulated after osimertinib treatment. However SK-216 did not increase (or reverse) E-cad mRNA. The reviewer wondered whether SK-216 really inhibits EMT. Please discuss this issue.
4. In Figure 4, expression of integrin-initiated EMT-related genes is shown to be upregulated in response to EGFR-TKI and its change is attenuated by PAI-1 inhibitor. Whereas this is interesting data, the reviewer wondered which signaling of PAI-1 or ECM change (tumor microenvironment) is initiation of tolerance mechanism to EGFR-TKI. How does PAI-1 increase αvβ3-integrin expression in the osimertinib tolerant cells? In addition, increased αvβ3 expression is reported to correlate with increased TGFβ signaling (doi: 10.1074/jbc.M109.018804). Please discuss.
5. In Figure 6, data of patients’ samples are really valuable. T790M status after 1st EGFR-TKI treatment should be indicated in the Table. In addition, all patients received EGFR-TKI other than osimertinib. Did they receive osimertinib treatment as 2nd EGFR-TKI treatment?
Author Response
Arasada et al. reported that serum PAI-1 expression is low before EGFR-TKI, and high after treatment was evaluated (DOI: 10.1038/s41467-018-05626-2). They also described that patients expressing low levels of pre-treatment PAI-1 showed an increase in PAI-1 levels after EGFR TKI therapy and have a decreased progression free-survival. Please cite this paper and discuss this issue.
Response
Thank you for your suggestion. Considering the importance of the issue you have pointed out, we have cited the paper and added the relevant information to the discussion section of our revised manuscript as follows (page 16, lines 586–596):
Since we have shown that PAI-1 is involved in EGFR-TKI tolerance, serum PAI-1 levels might be a useful predictor of resistance to EGFR-TKI and the therapeutic response of EGFR-TKI. According to our review, one previous study reported that serum PAI-1 level prior to EGFR-TKI treatment could be a surrogate marker for the therapeutic effect of EGFR-TKI [52]. This research shows that EGFR mutation-positive cells express PAI-1 through the Notch3-dependent β-catenin signaling induced by EGFR-TKI. Actually, in cases with lower pre-treatment serum PAI-1 levels, the serum PAI-1 levels at the time of resistance tended to increase. In addition, this study showed that lower pre-treatment serum PAI-1 levels are associated with poorer response to EGFR-TKI. However, pro-spective studies with a larger number of patients are needed to determine whether serum PAI-1 levels can act as a surrogate marker of EGFR-TKI treatment efficacy.
The label of Figure 1E is incorrect; osimertinib-tolerant PC-9 cells. In figure 1F, “gefitinib” means “samples from patients who were resistant to or after treatment of gefitinib?
Response
As you pointed out, Figure 1E was incorrectly described. We have corrected the caption in the revised paper.
Figure 1F is an experiment using cells obtained from the pleural effusion of the patient seven months after treatment with afatinib. The expression of PAI-1 mRNA was shown after gefitinib and osimertinib administration, in addition to other experiments using H1975 and PC-9. These descriptions have been added to the Figure 1 legend in our revised manuscript (page 7, lines 292–297, This data has been updated to Figure I within the new manuscript.).
In Figure 3, the mRNA expression of EMT markers as well as PAI-1 was significantly upregulated after osimertinib treatment. However SK-216 did not increase (or reverse) E-cad mRNA. The reviewer wondered whether SK-216 really inhibits EMT. Please discuss this issue.
Response
Thank you for your reasonable and important question. As you noted, E-cadherin was not upregulated after SK-216 administration, and rather tended to decrease. Certainly, this finding may not be consistent with the typical EMT process.Actually, additional time might be required for the elevation of E-cadherin levels. Recently, the concept of partial EMT, which indicate a transitional or intermediate state between epithelial and mesenchymal properties, has been proposed (Nat Rev Mol Cell Biol. 2020;21;341-352. DOI: 10.1038/s41580-020-0237-9). As shown in Figure 3, although mesenchymal markers were elevated after osimertinib administration, E-cadherin expression was not significantly changed. In addition, E-cadherin expression was not elevated after PAI-1 inhibition. We suggest that the osimertinib-tolerant cells showed partial EMT, in which only mesenchymal markers changed rapidly, and epithelial markers did not respond well.
This concept has been described in the discussion section of our revised manuscript (page 15, lines 541–547).
In Figure 4, expression of integrin-initiated EMT-related genes is shown to be upregulated in response to EGFR-TKI and its change is attenuated by PAI-1 inhibitor. Whereas this is interesting data, the reviewer wondered which signaling of PAI-1 or ECM change (tumor microenvironment) is initiation of tolerance mechanism to EGFR-TKI. How does PAI-1 increase αvβ3-integrin expression in the osimertinib tolerant cells? In addition, increased αvβ3 expression is reported to correlate with increased TGFβ signaling (doi: 10.1074/jbc.M109.018804). Please discuss.
Response
The research paper you presented is important for considering the relationship between integrins and PAI-1. However, since the findings in the paper did not explain the results of our research, we have not referred to them.
PAI-1 appears to induce EMT signaling by upregulating the expression of integrins through the following mechanism. The results of our microarray showed that osimer-tinib-tolerant cells expressed extracellular matrix (ECM) components such as vitronectin, fibronectin, laminin, and collagen. In addition, PAI-1, which is expressed by osimer-tinib-tolerant cells, can protect the ECM from proteolysis by inhibiting the local plas-minogen activator, resulting in the accumulation of ECM [47]. Subsequently, ECM-to-cell adhesion results in the formation of focal adhesions, which contain integrin, and increases the expression of integrin [48]. As described above, we propose that PAI-1 induces the accumulation of ECM, and then the expression of integrin is enhanced, eventually leading to osimertinib tolerance. However, further studies are required to elucidate the detailed molecular mechanism involved in the process.
These suggestions have been described in the discussion section of our revised manuscript (page 15, lines 565–574). The original sentences regarding the relationship between PAI-1 and integrins have been deleted.
In Figure 6, data of patients’ samples are really valuable. T790M status after 1st EGFR-TKI treatment should be indicated in the Table. In addition, all patients received EGFR-TKI other than osimertinib. Did they receive osimertinib treatment as 2nd EGFR-TKI treatment?
Response
As you noted, the presence of the T790M mutation is important. We have modified Table 1 to include the relevant data: The T790M mutation was identified in 3 out of 5 patients, all of whom received osimertinib as second-line therapy.

Reviewer 2 Report (New Reviewer)
Thank you for your revision of the article “Association between plasminogen activator inhibitor-1 and osimertinib tolerance in EGFR-mutated lung cancer via epithelial–mesenchymal transition”. The revisions has improved the article. However, the following points need to be addressed to further improve the quality of this manuscript.
Major comments:
1. Authors have added the siRNA-based PAI-1 knockdown experiment, but there are several concerns about this experiment. First, at least two different PAI-1 specific siRNAs have to be used to avoid off-target effect. Second, western blotting should be done to show the siRNA knockdown efficiency. Third, the results of siRNA knockdown on the proliferation of EGFR-TKI-tolerant cells look weird (Fig2F&G) because even the NC-siRNA led to the decrease of cell numbers. For my experience, the toxicity of Lipofectamine RNAiMAX transfection reagent is not high.
2. Authors only performed microarray analysis and qRT-PCR to address the association of osimertinib tolerance and EMT. These results are not enough to support the conclusion “PAI-1 to be involved in tolerance to osimertinib via EMT”. At least invasion and migration assays should be conducted.
3. Authors only performed ELISA to show the PAI-1 protein levels, but did not show the PAI-1 protein levels in cells by western blotting.
4. Since it has been reported that EMT is associated with osimertinib resistance mechanism, it should be described in the Introduction.
Minor comments:
Page 5, line 179: “H19675” should be “H1975”
Page 11, line 392: The sentence of “To confirm the increase in PAI-1 expression in EGFR-TKI-tolerant cells in vivo, we used H1975 cells, in which EMT signaling was strongly suppressed by PAI-1 inhibition in vitro.” looks not complete. we used H1975 cells to do what?
Page 13, line 450: “Figure S6” should be “Figure S7”.
The labels of (A) and (B) are missing in the Figure S2, S5 and S6.
Figure S5: “TGF-b” should be “TGF-β”.
Author Response
Authors have added the siRNA-based PAI-1 knockdown experiment, but there are several concerns about this experiment. First, at least two different PAI-1 specific siRNAs have to be used to avoid off-target effect. Second, western blotting should be done to show the siRNA knockdown efficiency. Third, the results of siRNA knockdown on the proliferation of EGFR-TKI-tolerant cells look weird (Fig2F&G) because even the NC-siRNA led to the decrease of cell numbers. For my experience, the toxicity of Lipofectamine RNAiMAX transfection reagent is not high.
Response
Thank you for your important suggestions. We understand the need to use two siRNAs. Therefore, we performed the experiment using different PAI-1 siRNA (ID: s10014) and showed that the results were similar.
Regarding your second comment, we performed western blotting using the cells after siRNA treatment to confirm that knockdown by PAI-1-siRNA reduced the expression of the PAI-1 protein, as well as mRNA. However, because of the small number of cells and lack of proliferation, sufficient protein was not obtained and could not evaluated by western blotting. Therefore, we only mentioned that we confirmed the knockdown by PCR this time, as in the first manuscript (page 8, lines 311–313).
As for the third point, the reason for the decreasing cell number treated with NC-siRNA might be the effect of exposure to serum-free culture medium. As shown in Figure 1C and D, PAI-1 is also highly accumulated in the culture medium. To avoid this effect, the culture medium was replaced with serum-free culture medium immediately before siRNA knockdown. We have added this explanation to the materials and methods section of our revised manuscript (page 5, lines 204–205).
Authors only performed microarray analysis and qRT-PCR to address the association of osimertinib tolerance and EMT. These results are not enough to support the conclusion “PAI-1 to be involved in tolerance to osimertinib via EMT”. At least invasion and migration assays should be conducted.
Response
We agree that evaluating the properties of EMT using invasion and migration assays is important. Thus, we have compared the degree of migration and invasion between EGFR-TKI-tolerant, developed EMT, and control cells using the QCM Chemotaxis Cell Migration Assay kit (MERCK) and CytoSelect 24-well Cell Invasion Assay kit (Cell Biolabs, Inc.). Unfortunately, this experiment did not show a significant difference in the degree of migration and invasionbetween the cells. This result may be due to the fact that, mesenchymal cells tended to proliferate slower than controlcells. As suggested by the reviewer, we planned to present the comparison of EMT properties between mesenchymal cells and control cells; however, we found it difficult.
Authors only performed ELISA to show the PAI-1 protein levels, but did not show the PAI-1 protein levels in cells by western blotting.
Response
Thank you for your suggestion. We have analyzed the expression of PAI-1 protein in cells by western blotting. We observed that PAI-1 protein in EGFR-TKI-tolerant cells was increased. Similarly, PAI-1 protein in the culture medium of EGFR-TKI-tolerant cells was increased. These data were described in the results section of our revised manuscript (page 7, lines 286–290).
Since it has been reported that EMT is associated with osimertinib resistanc mechanism, it should be described in the Introduction.
Response
We have mentioned several previous studies reporting the involvement of EMT in osimertinib resistance in the introduction section of our revised manuscript (page 3, lines 77–78).
Page 5, line 179: “H19675” should be “H1975”
Response
Thank you for pointing this out. This part has been corrected.
Page 11, line 392: The sentence of “To confirm the increase in PAI-1 expression in EGFR-TKI-tolerant cells in vivo, we used H1975 cells, in which EMT signaling was strongly suppressed by PAI-1 inhibition in vitro.” looks not complete. we used H1975 cells to do what?
Response
We intended to explain that we performed animal studies with H1975, in which EMT signaling was strongly suppressed by PAI-1 inhibition in vitro. We have added a note to clarify this point (page 11, line 422).
Page 13, line 450: “Figure S6” should be “Figure S7”.
Response
Thank you for pointing this out. This part has been corrected.
The labels of (A) and (B) are missing in the Figure S2, S5 and S6.
Response
Thank you for pointing this out. This part has been corrected.
Figure S5: “TGF-b” should be “TGF-β”.
Response
Thank you for pointing this out. This part has been corrected.

Round 2
Reviewer 2 Report (New Reviewer)
Thank you for your revision of the article “Association between plasminogen activator inhibitor-1 and osimertinib tolerance in EGFR-mutated lung cancer via epithelial–mesenchymal transition”. The majority of revisions looks ok. However, I don’t understand the difference of the EGFR-TKI-tolerant and control cancer cells derived from patients (Figure 1I)? Which part of the patient did these two kinds of cells come from? How did isolate and culture these cells? I think the explanation should be included in the “Methods” sections.
Author Response
Thank you for your revision of the article “Association between plasminogen activator inhibitor-1 and osimertinib tolerance in EGFR-mutated lung cancer via epithelial–mesenchymal transition”. The majority of revisions looks ok. However, I don’t understand the difference of the EGFR-TKI-tolerant and control cancer cells derived from patients (Figure 1I)? Which part of the patient did these two kinds of cells come from? How did isolate and culture these cells? I think the explanation should be included in the “Methods” sections.
Response
Thank you for your feedback. We apologize for the confusing description. Our original description seemed to imply that the cancer cells were obtained from more than one patient. Actually, we obtained the cancer cells from only “one” patient. As described in the revised manuscript, this patient developed pleural dissemination after acquiring resistance to afatinib, and the T790M mutation was detected from cancer cells in the pleural effusion. The cancer cells were passaged and treated with gefitinib and osimertinib. These cells had the T790M mutation, which is resistant to gefitinib. Thus, most of these cells survived, and tolerance was not induced by gefitinib. Moreover, PAI-1 expression was not increased. In contrast, since osimertinib has an antitumor effect on T790M-positive cells, most cells treated with osimertinib died, and only tolerant cells survived and expressed high levels of PAI-1.
We have corrected the description and have added the necessary information in our revised manuscript. The corrected text is indicated in red. In addition, the procedure for isolating and culturing the cancer cells has been added to the revised supplementary methods.

This manuscript is a resubmission of an earlier submission. The following is a list of the peer review reports and author responses from that submission.
Round 1
Reviewer 1 Report
The authors presented a role of plasminogen activator inhibitor-1 (PAI-1) in osimertinib tolerance in EGFR-mutated noon-small cell lung cancer (NSCLC) via epithelial–mesenchymal transition (EMT).
They showed that the expression of PAI-1 and mesenchymal genes in EGFR-mutated NSCLC cell lines was upregulated with tolerance to EGFR-TKIs in vitro, and that PAI-1 inhibition using inhibitor, SK-216 attenuated the proliferation and mesenchymal gene expression of EGFR-TKI-tolerant cells.
NSCLC patients develop secondary resistance, which constitutes a critical challenge due to the scarcity of post-osimertinib pharmacological options available. Thus this manuscript is logical and interesting, and discussed a hot topic in NSCLC treatment.
However, the following points should be addressed.
The author should discuss as follows:
(1) In this study, the authors used SK-216, a specific PAI-1 inhibitor, to evaluate the role of PAI-1 on the proliferation and mesenchymal gene expression of EGFR-TKI-tolerant cells. Whereas the result was interesting, could the authors use the result using knockdown or overexpression model in order to show the direct effect of PAI-1 on EGFR-TKI tolerance acquisition?
(2) SK-216 did not attenuate change of PAI-1 or E-cadherin expression after osimertinib treatment in NSCLC cells (Figure 3). How did the compound make effect on cell behavior? Did PAI-1 protein expression decrease in SK-216 treated cells? In addition, microarray analysis was important to understand the signaling pathway between PAI-1 and EMT as well as the mechanism of PAI-1 for attenuation of EGFR-TKI tolerance acquisition (Figure 4A-B). Thus genetic analysis between osimertinib treated cells and osimertinib+SK-216 treated cells.
(3) Please show the PAI-1 immunostaining result in tumor from mice treated with osimertinib+SK-216 (Figure 5E-F). Quantitative PAI-1 expression should be shown by qRT-PCR or western blotting using tumors from each group.
(4) In Table 1, Gefitinib, Erlotinib, and Afatinib were used for EGFR-TKI treatment. Was mutation status evaluated after EGFR-TKI treatment? The reviewer wondered whether additional genetic mutation related EGFR-TKI tolerance was not associated with PAI-1 induced EGFR-TKI tolerance acquisition.
(5) In this study, upregulation of PAI-1 expression is a key factor in EGFR-TKI tolerance acquisition, thus the mechanism between EGFR-TKI treatment and PAI-1 expression change should be discussed.
Is it possible to evaluate PAI-1 expression in patient-derived serum samples? The reviewer wondered whether PAI-1 may be one of surrogate biomarkers for EGFR TKI treatment.
Reviewer 2 Report
The authors indicate that PAI-1 is involved in osimertinib resistance via EMT in EGFR-mutated cancer cells.
The main message is very interesting, however the manuscript needs robust improvement. Please see my comments below.
Tolerance is inappropriately used: if the author intend resistance to the drug explored here, please replace with resistant.
How did the authors distinguish between TKI-tolerant vs resistant cells?
Line 207: The concentration of SK-216 should be expressed uniformly throughout the text
Fig. 1: 1B-D-F need One Way ANOVA; 1E: indicate in the figure the type of cells for better understanding
Fig. 1: RT-PCR experiments: the authors indicate mean fold changes in the legend, but the graphs have the ratio of the target vs b-actin on the y axis. Please explain
Fig. 1: it is not clear whether the cells were treated or not: very confusing! If they were treated, the concentration of the drugs need to be indicates and the vehicle control is mandatory.
Fig. 2: the treatment is not clear. Specify, otherwise it is difficult to interpret the data.
Line 268: “In addition, PC-9 cells were not found to harbor any T790M or C797S mutations after gefitinib or osimertinib treatment. In H1975 cells harboring T790M mutation, C797S was not observed after osimertinib treatment.” This statement is not corroborated by evidence.
Fig. 3: it is not clear the difference between Fig. 3A-B vs C-D. Better specify. Again, the y axis for RT-PCR data refers to the ratio of the target gene vs b-actin and not to fold changes as stated in the legend. The same is in Fig. 4 C-F.
Fig. 5: what solvent is the control for Osimertinib? Students’ t test is not appropriate. Fig. 5B: indicate the treatment in the figure for better understanding
Fig. 5C: what OR corresponds to? It is very confusing! The other figures have OS and then OT. The positive area to PAI-1 is not convincing.
Fig. 5D: Students’ t test is not appropriate
Fig. 6: 3 staining for PAI-1 is not enough. The number of specimens need to be increased. Moreover, the three patients show a high variability in PAI-1 staining. For example, patient 1 seems to have much more PAI-1 than patient 3, that the authors equally considered as positive. Patient 2 is an intermediate between patient 1 and 3. Therefore, more human samples are needed to corroborate the message the authors want to send.